# Correlation-based tests for the formal comparison of polygenic scores in multiple populations

**Sophia Gunn**[¤]*****, **Kathryn L. Lunetta**

Department of Biostatistics, Boston University School of Public Health, Boston, Massachusetts, United States of America

¤ Current address: New York Genome Center, New York, New York, United States of America
* sgunn@nygenome.org

## Abstract

Polygenic scores (PGS) are measures of genetic risk, derived from the results of genome wide association studies (GWAS). Previous work has proposed the coefficient of determination ($R^2$) as an appropriate measure by which to compare PGS performance in a validation dataset. Here we propose correlation-based methods for evaluating PGS performance by adapting previous work which produced a statistical framework and robust test statistics for the comparison of multiple correlation measures in multiple populations. This flexible framework can be extended to a wider variety of hypothesis tests than currently available methods. We assess our proposed method in simulation and demonstrate its utility with two examples, assessing previously developed PGS for low-density lipoprotein cholesterol and height in multiple populations in the All of Us cohort. Finally, we provide an R package 'coranova' with both parametric and nonparametric implementations of the described methods.

**Data Availability Statement:** The PGS for LDL cholesterol and height can be downloaded from the PGS catalog with accession numbers PGP000230 and PGP000382, respectively. The coranova R package can be downloaded from github (https://

## Author summary

Polygenic scores (PGS) are measures of genetic risk of disease that have been widely embraced by the scientific community. While there are many methods available to develop PGS, we have limited tools by which to compare PGS performance. Previous work has proposed an $R^2$-based approach which appropriately accounts for the correlation between PGS when comparing their performance. Here, we propose correlation-based tests which can assess multiple scores in multiple populations while accounting for the correlation between the scores. Our method is highly flexible and can be used by researchers to test any linear hypothesis of PGS performance, though we suggest three ANOVA-like tests as a starting point. We apply our method to PGS developed for LDL cholesterol and height in the All of Us cohort. In these examples, we demonstrate how our method can be used by researchers to compare and evaluate PGS in multiple populations. This approach will be particularly useful as we look to improve PGS performance in

github.com/gunns2/coranova). This study used data from the All of Us Research Program's Controlled Tier Dataset V6, available to authorized users on the Researcher Workbench. Instructions for access to the All of Us Researcher Workbench is available at https://www.researchallofus.org/register/. We have also included a jupityr notebook script that we used to compute the PGS in the AoU population in the coranova github page, within the folder "analysis".

**Funding:** SG and KLL were supported by funding from the National Heart Lung and Blood Institute (SG: F31HL163952; KLL: R01HL092577). SG was additionally supported by National Institute of General Medical Sciences (5T32GM074905-15). The funders had no role in study design, data collection and analysis, decision to publish, or preparation of the manuscript.

**Competing interests:** The authors have declared that no competing interests exist.

underrepresented populations in genetic research and need to evaluate PGS in multiple populations to appropriately assess PGS performance.

## Introduction

The rise of large genome wide association studies (GWAS) has enabled researchers to build models for individual genetic risk prediction, called polygenic scores (PGS) [1]. Polygenic scores predict genetic risk for a given trait with a weighted sum of relevant risk alleles. The risk allele weights are derived from GWAS effect estimates. There are many different methods available for PGS development. At minimum, PGS methods require a GWAS from which to derive the weights, and most also require a linkage disequilibrium (LD) reference and training data to optimize parameters [2]. Thus, for any given trait, many different polygenic scores can be derived. Further, the PGS once developed can also be applied to different populations, with varying performance due to factors like differences in allele frequencies and LD patterns [3]. With many possible PGS to choose from, there is a need to develop methods to assess and compare the performance of polygenic scores.

A popular measure of assessment for polygenic scores is the $R^2$ from a linear regression model fit in a validation dataset with the PGS as the primary predictor and the trait of interest as the dependent variable, adjusting for relevant covariates. The $R^2$ of a linear regression model is the proportion of variance explained by covariates in the model, also called the coefficient of determination [4]. Thus, the $R^2$ from such a model can be interpreted as the proportion of variance explained by the PGS. This approach is appealing because of the connection to heritability. The heritability of a trait is the proportion of phenotypic variation that is explained by additive genetic variation [5]. The $R^2$ of a polygenic score is limited by the snp-based heritability, or the proportion of phenotypic variation that is explained by SNPs, of its associated trait. The closer the $R^2$ of a proposed PGS is to the snp-based heritability, the better the score [2, 6].

Momin et al. proposed a formal statistical framework for comparing polygenic scores with $R^2$, called R2 Redux [7]. Using results from Olkin & Finn to generate asymptotic distributions of $R^2$, they devised methods to compute the variance and generate confidence intervals for the difference between the $R^2$ of two polygenic scores [8]. They also proposed methods for determining the difference in $R^2$ for nested models, for independent groups, and for genomic partitioning analysis.

While Momin et al.'s $R^2$-based approach [7] is ideal in the applications for which it was designed, R2Redux was not designed for testing multiple scores in multiple populations simultaneously. To address this gap, we propose using correlation-based methods to assess the performance of polygenic scores by adapting the work of Olkin & Finn [9] and Bilker et al. [10]. Olkin & Finn derived the asymptotic joint distribution of sample correlations between continuous predictors and continuous outcomes, when the predictors themselves are correlated, like in the case of polygenic scores. Further, they demonstrate how to derive linear hypothesis tests of the correlation measures. Bilker et al. adapted this work of Olkin & Finn and proposed an ANOVA-like testing framework for assessing correlation, called *Coranova*. They proposed specific hypothesis tests researchers can perform on correlated predictors in multiple independent population samples, applying the method to neurological exams.

We can use the Coranova framework to compare multiple PGS in multiple populations with the three Coranova hypothesis tests. We can assess whether the scores have the same correlation with the outcome of interest within population samples, whether the mean score correlation with the outcome differs between population sample and finally whether the pattern

of score performance differs by population sample. The Coranova hypothesis tests are an ideal starting point for researchers to analyze the performance of multiple PGS in multiple populations. However, crucially, researchers can also devise contrast matrices to implement correlation-based hypothesis tests specific to their research interests.

Correlation measures the linear relationship between two random variables, ranging between -1 and 1. For two random variables X and Y, their correlation is equal to their covariance divided by their standard deviations ($corr(X, Y) = \frac{cov(X,Y)}{sd(X)sd(Y)}$). In the case of a polygenic score and its associated outcome, this correlation is typically positive, ranging from 0 (the PGS is completely independent of its outcome), to 1 (the PGS is perfectly linearly associated with its outcome). PGS that have higher correlations with their outcomes are better and will be more predictive of their outcomes. The tests we introduce here are designed to compare the correlations of multiple polygenic scores and their associated outcome in multiple populations.

To demonstrate the connection between using correlation to evaluate PGS and more commonly used metrics, we consider a linear model,

$$Y = X\beta + \epsilon \tag{1}$$

with a single polygenic score, **X**, the quantitative, normally distributed trait that it predicts, **Y**, and $\epsilon$ is the vector of residuals. We assume both outcome and PGS have been adjusted for appropriate factors, such as genetic principal components. We further assume outcome and PGS have been standardized to mean 0 and standard deviation 1. The regression model will yield the estimate of $\beta$, $\hat{\beta} = \frac{\sum (x_i - \bar{x})(y_i - \bar{y})}{\sum (x_i - \bar{x})^2}$ which is equal to $\frac{cov(x,y)}{var(x)}$. The sample correlation $r_{xy} = \frac{\sum (x_i - \bar{x})(y_i - \bar{y})}{\sqrt{\sum (x_i - \bar{x})^2 \sum (y_i - \bar{y})^2}}$, is equal to $\frac{cov(x,y)}{sd(x)sd(y)}$. Rearranging we can see that $\hat{\beta} = r_{xy} \frac{sd(y)}{sd(x)}$, and thus in the case of standardized $Y$ and $X$, $\hat{\beta} = r_{xy}$. Further, in this case $r_{xy}^2$ is equal to the coefficient of determination ($R^2$) of the linear model (1). Thus, while the correlation-based tests we introduce here are not the same as $R^2$-based tests, they will generally lead to the same conclusions as $R^2$-based tests when performing pair-wise comparisons in samples of size 1000 or larger. Importantly, correlation-based tests also provide the opportunity to test more complex hypotheses about the performance of polygenic scores in multiple populations.

We have built an R package with both parametric and nonparametric implementations of the methods proposed by Olkin & Finn and Bilker et al. for polygenic score evaluation and comparison. With simulations we show that our correlation-based tests have well-controlled type 1 error rates and power greater than 80% to detect differences in polygenic score performance in multiple populations at typical sample sizes under reasonable assumptions of parameter values. Finally, we demonstrate our proposed methods with two real world applications to polygenic scores for low-density lipoprotein (LDL) cholesterol and height in the All of Us cohort (AoU) and provide examples for researchers interested in applying our methods.

## Description of the method

### Correlation-based tests

To define our correlation-based tests, we will first define our parameters. Let $\rho_{i,j}$ be the population Pearson correlation of PGS $i$ with outcome $Y$ in population $j$. Let $\mu$ denote the vector of these population correlations of $P$ PGS with outcome $Y$ in population j, $\mu = (\rho_{1,j}, \rho_{2,j}, \ldots \rho_{P,j})$ and let $u$ be the vector of sample estimates of $\mu$, $u = (r_{1,j}, r_{2,j}, \ldots r_{P,j})$.

Olkin & Finn defined the asymptotic covariance matrix of $u$, $\Sigma_{\infty}(u)$ (see Section A in S1 Appendix), and thus we can model $u - \mu$ as a multivariate normal random variable: $u - \mu \sim N(0, \Sigma_{\infty}(u))$. Further, they argued with this distribution defined, we can construct hypothesis

tests using contrast matrix $A$, where $A$ is a $m$ x $P$ matrix of rank $m$ where $H_0 : A\mu = \mu_0$. Then, the test-statistic $S$ is a $\chi^2$ random variable with $m$ degrees of freedom:

$$S = (Au - \mu_0)'(A\hat{\Sigma}_\infty(u)A')^{-1}(Au - \mu_0) \sim \chi^2_{df=m} \tag{2}$$

The covariance matrix of the sample correlations $u$, $\hat{\Sigma}(u)$, can be also be estimated by bootstrapping as suggested by Bilker et al. We have provided the option to use either the parametric derivation by Olkin & Finn or bootstrapping in our R package. To compare the correlation of $P$ PGS in $K$ populations, we can define $u = (r_{1,1}, r_{2,1}, \ldots r_{P,1}, r_{1,2}, r_{2,2}, \ldots r_{P,2}, \ldots, r_{1,K}, r_{2,K}, \ldots r_{P,K})$. Under the parametric derivation of $\hat{\Sigma}_\infty(u)$, we assume that the correlations are independent across populations. The bootstrap derivation of $\hat{\Sigma}(u)$ does not make this assumption.

## Coranova hypotheses

Bilker et al. introduced three types of hypothesis tests which enable ANOVA-like testing of correlated variables, like polygenic scores, in multiple population groups by specifying three contrast matrices that can be used in Eq (2).

Suppose we have $K$ population samples, and $P$ PGS for a given continuous trait $Y$. We are interested in comparing the correlations between the $P$ PGS and the trait $Y$ among the $K$ population samples. Specifically, we can 1) test for differences in the correlations between the trait $Y$ and $P$ PGS 2) test for differences in the associations between trait $Y$ and the $P$ PGS between the $K$ population samples, and 3) test for an interaction effect between the scores and populations, or in other words, test for differences within the pattern of correlations in the $K$ population samples.

Let $\rho_{i,j}$ be the population correlation between PGS $i$ in population $j$. We have $P$ PGS and $K$ population samples. The three hypothesis tests can be written as follows;

**Test for a within effect.** To test for differences in the correlations between trait $Y$ and the $P$ PGS within the $K$ population samples, let

$H_0 : \text{mean}(\rho_{i,1}, \ldots, \rho_{i,k})$ are equal for all $i \in 1, \ldots, P$

$H_A : \text{mean}(\rho_{i,1}, \ldots, \rho_{i,k}) \neq \text{mean}(\rho_{m,1}, \ldots, \rho_{m,k})$ for at least one $(i, m)$ pair of measures.

**Test for a between effect.** To test for differences in the correlations between the trait $Y$ and $P$ PGSs between the $K$ population samples, let

$H_0 : \text{mean}(\rho_{1,j}, \ldots, \rho_{P,j})$ are equal for all $j \in 1, \ldots, K$

$H_A : \text{mean}(\rho_{1,j}, \ldots, \rho_{P,j}) \neq \text{mean}(\rho_{1,l}, \ldots, \rho_{P,l})$ for at least one $(j, l)$ pair of populations.

**Test for an interaction effect.** To test for a difference in the pattern of correlations between the trait $Y$ and $P$ PGSs in the $K$ population samples, let

$H_0 : (\rho_{1,j} - \rho_{i,j}) - (\rho_{1,l} - \rho_{i,l}) = 0$ for all $i = 1, \ldots, P$ and $j, l \in 1, \ldots, k, j \neq l$

$H_A :$ at least one interaction not equal to 0.

Using the contrast matrices provided by Bilker et al. (with a typo corrected, see Section B in S1 Appendix), we can conduct these tests by generating $\chi^2$ test statistics following Eq (2).

## Verification and comparison

### Description of simulations

To assess our implementation of these correlation-based hypothesis tests we performed simulations with K = 2 independent population samples with $p$ polygenic scores built with different methods $(X_1, X_2, \ldots, X_p)$ and a continuous outcome $Y$. We simulated the outcome and polygenic scores using a multivariate normal distribution for each population, with a specified covariance matrix and sample size $n$.

All variables were simulated as standardized normal random variables with mean 0 and standard deviation 1. In the case of three polygenic scores ($p = 3$), the general correlation

structure for population sample $K \in A, B$ of the outcome $Y_K$ and three polygenic scores $X_{K1}$, $X_{K2}, X_{K3}$ is as follows,

$$Corr(Z_K) = Corr(Y_K, X_{K1}, X_{K2}, X_{K3}) = \begin{bmatrix} 1 & \tau_1 & \tau_2 & \tau_3 \\ \tau_1 & 1 & \phi & \phi \\ \tau_2 & \phi & 1 & \phi \\ \tau_3 & \phi & \phi & 1 \end{bmatrix}$$

For each simulated population sample, $\tau_i$ is the correlation of PGS $i$ with the outcome and $\phi$ is the common inter-PGS correlation.

We used the following correlation matrices to generate simulated data to assess our methods with two populations and three PGS, where correlation matrix $Corr(Z_A)$ is used to generate the first population sample and correlation matrix $Corr(Z_B)$ is used to generate the second population sample.

I. To simulate a difference across polygenic scores within the groups we used the following correlation matrices:

$$Corr(Z_A) = Corr(Z_B) = \begin{bmatrix} 1 & \tau & \tau & \tau + \delta \\ \tau & 1 & \phi & \phi \\ \tau & \phi & 1 & \phi \\ \tau + \delta & \phi & \phi & 1 \end{bmatrix}$$

Under this setting, in both populations the third polygenic score has correlation with the outcome $Y$ of $\tau + \delta$ in comparison with the first two polygenic scores which have the same correlation with $Y$, $\tau$. Thus, when $\delta > 0$, the null hypothesis of equal correlation of the scores within the groups does not hold.

II. To simulate a difference across groups we used the following correlation matrices:

$$Corr(Z_A) = \begin{bmatrix} 1 & \tau & \tau & \tau \\ \tau & 1 & \phi & \phi \\ \tau & \phi & 1 & \phi \\ \tau & \phi & \phi & 1 \end{bmatrix}, Corr(Z_B) = \begin{bmatrix} 1 & \tau + \delta & \tau + \delta & \tau + \delta \\ \tau + \delta & 1 & \phi & \phi \\ \tau + \delta & \phi & 1 & \phi \\ \tau + \delta & \phi & \phi & 1 \end{bmatrix}$$

Under this setting, all three polygenic scores in population sample A have correlation $\tau$ with the outcome $Y$, while all three polygenic scores in population sample B have correlation $\tau + \delta$ with the outcome $Y$. Thus, when $\delta > 0$, the null hypothesis of equal correlation of the scores with the outcome between the populations does not hold.

III. Finally, to simulate a difference across groups and across polygenic scores with two samples we used the following correlation matrices:

$$Corr(Z_A) = \begin{bmatrix} 1 & \tau & \tau & \tau \\ \tau & 1 & \phi & \phi \\ \tau & \phi & 1 & \phi \\ \tau & \phi & \phi & 1 \end{bmatrix}, Corr(Z_B) = \begin{bmatrix} 1 & \tau & \tau & \tau + \delta \\ \tau & 1 & \phi & \phi \\ \tau & \phi & 1 & \phi \\ \tau + \delta & \phi & \phi & 1 \end{bmatrix}$$

Under this setting, all three polygenic scores in population sample A have correlation $\tau$ with the outcome $Y$, while the first two polygenic scores in population sample B have correlation $\tau$ with the outcome $Y$ and the third polygenic score in population sample B has correlation $\tau + \delta$ with $Y$. Thus, when $\delta > 0$, the null hypothesis of no difference in pattern of score performance

across the groups does not hold, and there is an interaction between the polygenic scores and population samples.

To assess the performance of the methods with multiple population samples and polygenic scores, we simulated one thousand replicates for each hypothesis and combination of $\tau$, $\phi$, $\delta$, $n$. We simulated $\tau$ levels of 0.05, 0.1, 0.2, 0.4, and 0.6, $\delta$ levels of 0, 0.01, 0.03, 0.05, 0.075 and 0.1, and $\phi$ levels of 0.3, 0.5, 0.7 and 0.9. We simulated population samples of 500, 1000 and 5000, and assessed the performance our methods under equal and unequal sample sizes.

We also simulated data with a single population and two PGS to compare our method to the method proposed by Momin et al. with $R^2$ Redux [7]. Performing the Coranova "within" based hypothesis test on one population with two scores is identical to testing the difference in correlation between two scores. We simulated $\tau$ levels of 0.05, 0.1, 0.2, 0.4, 0.6 and 0.8, $\delta$ values of 0, 0.01, 0.03, 0.05, 0.075 and 0.1, and $\phi$ levels of 0.5, 0.7 and 0.9. We simulated population samples of 500, 1000 and 10000.

## Simulation results

When applied to two independent population samples and three polygenic scores, we find that the type I error rate of the proposed method is well controlled for all three Coranova hypotheses at alpha = 0.025 and 0.05 (Fig 1, Figs A-C in S3 Appendix).

As expected, the power of the three Coranova hypothesis tests increases when sample size increases, and when the magnitude of the difference between the scores' correlation with the outcome ($\delta$) increases (Fig 2, Figs D and E in S3 Appendix). We also see an increase in power

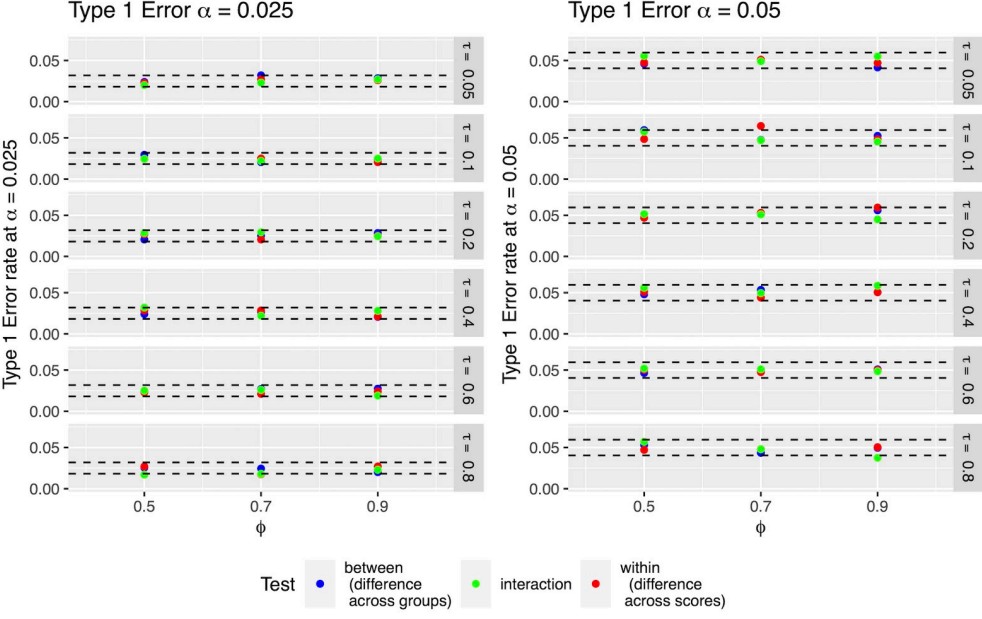

**Fig 1. Type I error of parametric implementation of Coranova applied to three PGS in two population samples of size N = 1000.** Each point represents the proportion of tests in 1000 simulations in which the null hypothesis was rejected at the specified alpha level with sample size 1000. Dashed lines indicate 95% confidence interval for specified alpha given sample size. In all simulations, the three scores have the same correlation with the outcome ($\tau$) in both population samples; $\phi$ is the correlation between the scores themselves within the population samples. In the "between" setting, the null hypothesis is that all the correlation of the scores and outcome is equal across population samples. In the "within" setting, the null hypothesis being tested is that the scores have the same correlation with the outcome within the population samples. In the "interaction" setting, the null hypothesis being tested is that the pattern of score performance is the same across the population samples.

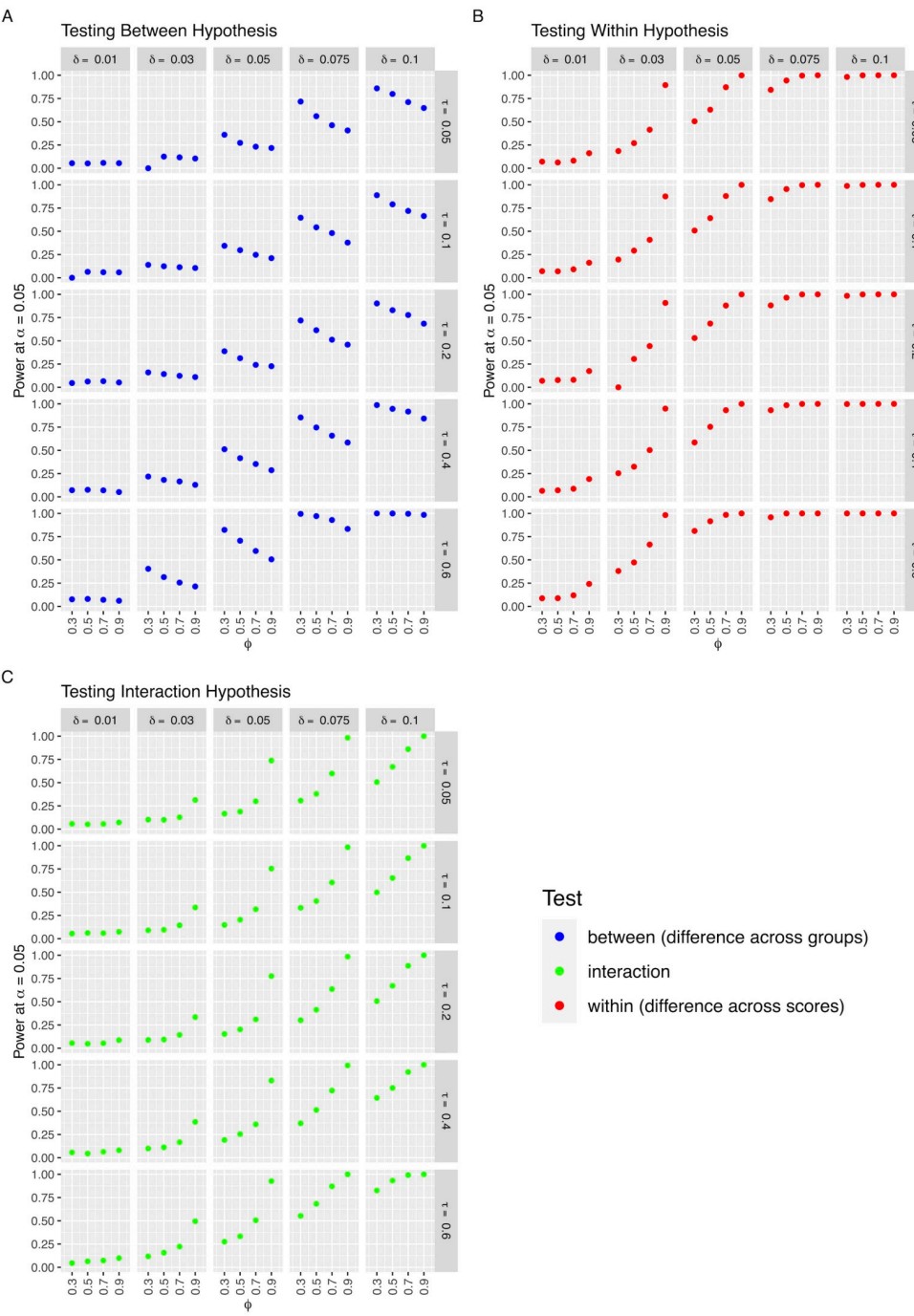

**Fig 2. Power of Coranova to detect a difference in performance between three PGS in two population samples of size N = 1000 under three simulation settings.** Each point represents the proportion of tests in 1000 simulations with parameters $\phi$, $\tau$ and $\delta$ in which the null hypothesis was rejected at significance level alpha = 0.05. A: *"Between" setting*, Within each population, the three PGS have equal correlation with outcome Y. This correlation differs between the two populations; $\tau$: correlation of each PGS with Y in population 1; $\tau + \delta$: correlation of each PGS with Y in population 2. B: *"Within" setting*, Each PGS has the same correlation with Y in the two populations, but the correlation of the third PGS with Y differs from the other two in the same way in both populations; $\tau$: correlation between first and second PGS with Y in both populations; $\tau + \delta$: correlation of 3rd PGS with Y in both populations. C: *"Interaction" setting*, One PGS differs from the other two in only one of the two populations; $\tau$: correlation of each of the three PGS with Y in population 1. Correlation of PGS 1 and 2 with Y in population 2; $\tau + \delta$: correlation of the third PGS with Y in population 2.

when the PGS are more correlated with the outcome (high $\tau$), due to the inverse relationship with the test-statistics and the variance terms of the correlations (see section A in S2 Appendix for further details).

When testing for differences in score performance between groups, power slightly decreases as correlation between scores increases (Fig 2A). When testing for differences in score performance within population groups, we see the opposite relationship; power increases as correlation between scores increases (Fig 2B). The power of the interaction test also increases with correlation between the scores (Fig 2C). We can understand the relationship between power and the correlation between scores by considering the covariance terms in the test-statistics. The between test-statistic ($\chi^2_{between}$) is inversely correlated with the covariance terms, while the within and interaction test-statistics ($\chi^2_{within}$, $\chi^2_{interaction}$) are positively correlated with the covariance terms. See section A in S2 Appendix for greater detail.

With a sample size of 1000 we find we have at least 80% power for the within hypothesis test when testing a difference in correlation of at least 0.075; for the between hypothesis test when testing a difference in correlation of at least 0.1 for scores with at least 0.4 correlation with the outcome; and for the interaction hypothesis when testing a difference in correlation of at least 0.1 for scores with inter-PGS correlation of 0.7 or higher. The power of the tests increases substantially with a sample size of 5000 (Fig E in S3 Appendix).

Briefly, we also found our correlation-based tests perform very similarly to R2 Redux when assessing the difference in performance between two polygenic scores in a single population (see section B in S2 Appendix, and Figs I-L in S3 Appendix).

## Applications

### Examples using All of Us cohort data

**Data.  Description of the All of Us (AoU) cohort**. All of Us is a diverse cohort made up of people living in the United States established by the National Institutes of Health [11]. Here we use the whole genome sequencing sample contained in All of Us Controlled Tier Dataset V6, released in June 2022. In addition to the quality control performed by the All of Us research team [12], we restricted the set of variants in this analysis to biallelic variants with a minor allele frequency greater than 0.001. We also restricted our sample to unrelated individuals.

The individuals in All of Us were grouped according to genetic similarity to the superpopulations in the Human Genome Diversity project (HGDP) [13] and 1000G samples (1KG) [14]. The AoU research team trained a random forest model on chromosomes 20 and 21 from HGSP and 1KG, and this model was applied to the AoU cohort to generate what we will call the 1KG genetic-similarity groups, in line with recommendations by the National Academies of Sciences, Engineering, and Medicine [15].

**Description of polygenic scores**. To demonstrate the utility of our proposed methods, we considered previously defined polygenic scores for two outcomes, LDL cholesterol and height. For both outcomes, we will compare PGS developed with GWAS of varying populations, corresponding to genetic-ancestry groups defined in the original papers. Using the language of the original papers, we will describe scores built with GWAS of a population that is genetically-similar to a single 1KG-superpopulation as an *ancestry-specific PGS*, and scores built with the GWAS results of a meta-analysis of multiple 1KG-superpopulations as *multi-ancestry PGS*.

We considered 12 polygenic scores for LDL cholesterol developed and made publicly available by Graham et al. [16]. These PGS were developed by The Global Lipids Genetics Consortium with data from 201 studies. The twelve polygenic scores were optimized using two methods: PRS-CS [17] and pruning and thresholding (PT) [2] with six GWAS performed on

**Table 1. Sample sizes of each 1KG genetic-similarity group available for each trait.**

|  | African | Admixed American | European | East Asian |
|---|---|---|---|---|
| LDL cholesterol | 2752 | 1073 | 7971 | NA |
| Height | 21184 | 14018 | 46948 | 2045 |

samples of varying genetic ancestry, as defined in the original paper. One score for each method is built using multi-ancestry meta-analysis GWAS results and the other five consist of ancestry-specific GWASs of samples from populations of African, East Asian, South Asian, European, and Hispanic ancestry. For the single-ancestry pruning and thresholding scores, a UK Biobank ancestry-matched sample was used to estimate LD, and for the multi-ancestry PT score, a mixed ancestry sample of the UKBB was used to estimate LD. For the single-ancestry PRS-CS scores, the LD reference panels were derived from ancestry-matched samples from 1000 Genomes [14]. A mixed ancestry sample of the 1000 Genomes was used to estimate LD for the multi-ancestry PRS-CS score. The polygenic scores were downloaded from the PGS catalog (publication ID: PGP000230).

For height, we considered six polygenic scores developed by Yengo et al, on behalf of the GIANT consortium [18]. All of Us participants were not included in the discovery sample. The scores were developed with SBayesR [19]. There are five scores built with ancestry-specific GWAS and one score built with a multi-ancestry meta-analysis GWAS. Each of the ancestry-specific scores was built with ancestry-group matched LD matrix, and the multi-ancestry score was built with LD estimated from a European population sample. The ancestry-specific scores correspond to European, African, Hispanic, South Asian, and East Asian populations as defined by the original paper. The scores were downloaded from the PGS catalog (publication ID: PGP000382).

**Score calculation.** Polygenic scores were computed in the AoU cohort using the PLINK2 [20] `--score` function. The phenotypes for both height and LDL cholesterol was determined by first computing the mean of the available measurements for each individual and then transforming the values by inverse-rank normalization.

We evaluated the polygenic scores in each AoU 1KG genetic-similarity group separately. Sample sizes of these genetic-similarity groups are included in Table 1. We computed the LDL cholesterol scores among the AoU individuals classified as similar to the African, Admixed American and European 1KG populations. We computed the height PGS among the AoU individuals classified as similar to the African, admixed American, European and East Asian 1KG populations. The phenotypes and PGS were adjusted for the first 10 ancestry principal components using linear regression within each genetic-similarity group prior to analysis using our correlation-based methods.

## Application results

**LDL Cholesterol examples.** The correlations between the polygenic scores for LDL cholesterol and inverse-rank mean LDL cholesterol in the African, admixed American, and European 1KG genetic-similarity groups in AoU are displayed in Fig 3. To analyze the performance of the PGS using our proposed methods, we first applied the three Coranova hypotheses to the 12 polygenic scores for LDL cholesterol across the three 1KG genetic-similarity groups. We find significant evidence that at least one score has higher correlation with LDL cholesterol than the others ($p_{within} = 1.9 \times e^{-71}$, Fig 3C), and significant evidence that the pattern of score correlation with LDL cholesterol differs across the 1KG genetic-similarity groups ($p_{interaction} =$

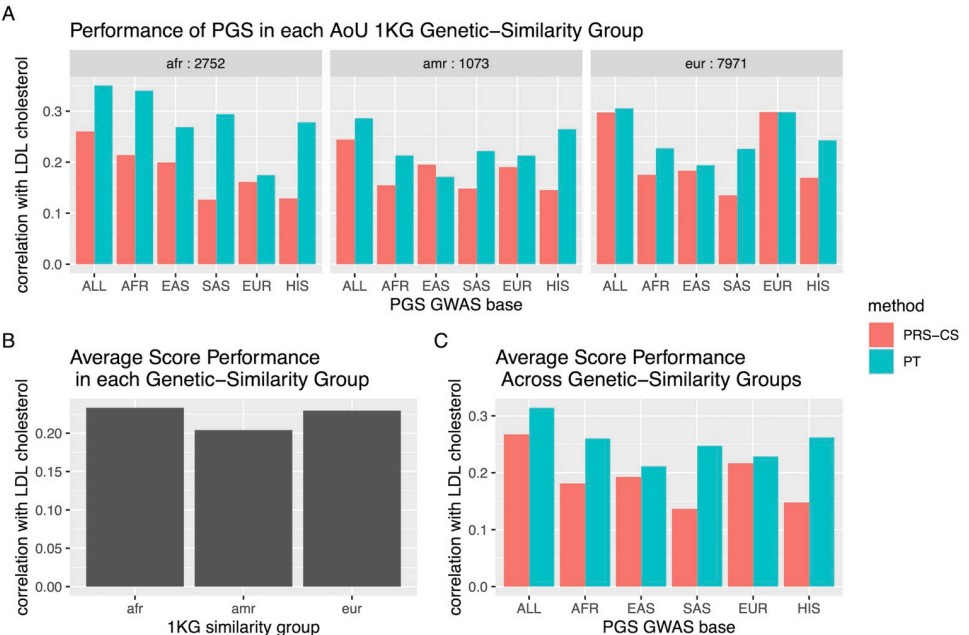

**Fig 3. Correlation of Graham et al. PGS with inverse rank normalized mean LDL cholesterol level in each of the African, Admixed American and European 1KG genetic-similarity groups in AoU.** A: The correlation with LDL cholesterol for each PGS by 1KG genetic-similarity group, where each PGS is identified by the GWAS base and method with which it was derived. Sample size of each group specified. B: The average score performance in each AoU 1KG similarity group, corresponding to the "between hypothesis test" in which the null hypothesis is that the scores perform on average the same in the three groups. C: The performance of each PGS averaged across the AoU 1KG similarity groups, corresponding to the "within hypothesis test" in which the null hypothesis is that the scores have the same correlation with the outcome when averaged across populations.

$1.5 \times e^{-33}$). We do not find significant evidence that the mean PGS correlation with LDL cholesterol differs across the 1KG genetic-similarity groups ($p_{between}$ = 0.4, Fig 3B).

In addition to the Coranova hypothesis tests, we can use the flexible framework to ask additional questions about the performance of the 12 PGS. One major question we can use the framework to assess is how the scores built with pruning and thresholding compare to the scores built with PRS-CS. We find that the PT PGS have higher correlation with LDL cholesterol than the PRS-CS scores, and at least one of the pairwise differences between the PT and PRS-CS scores built with the same GWAS significantly different from 0 ($p = 1 \times e^{-30}$, see section E in S1 Appendix). We can also compare the correlation of the multi-ancestry PT PGS to the ancestry-specific PT PGSs (Fig 4). Using a contrast matrix designed for this comparison, we fail to reject the null hypothesis that the multi-ancestry PGS and ancestry-specific scores have equal correlation with LDL ($p = 0.16$, see section E in S1 Appendix). Finally, we can also assess whether the multi-ancestry PT PGS differs in correlation with inverse-ranked mean LDL cholesterol across the three 1KG genetic-similarity groups using the Coranova 'between' hypothesis test on just the multi-ancestry PT PGS in the three groups. With this test, we find that the performance of the multi-ancestry PT PGS varies across the genetic-similarity groups ($p_{between}$ = 0.04).

**Height examples.** The correlations between the polygenic scores for height and inverse-rank mean height in the African, admixed American, European, and East Asian 1KG genetic-similarity groups in AoU are displayed in Fig 5. Considering the six scores in the four genetic-similarity groups, we find significant evidence that the average score correlation

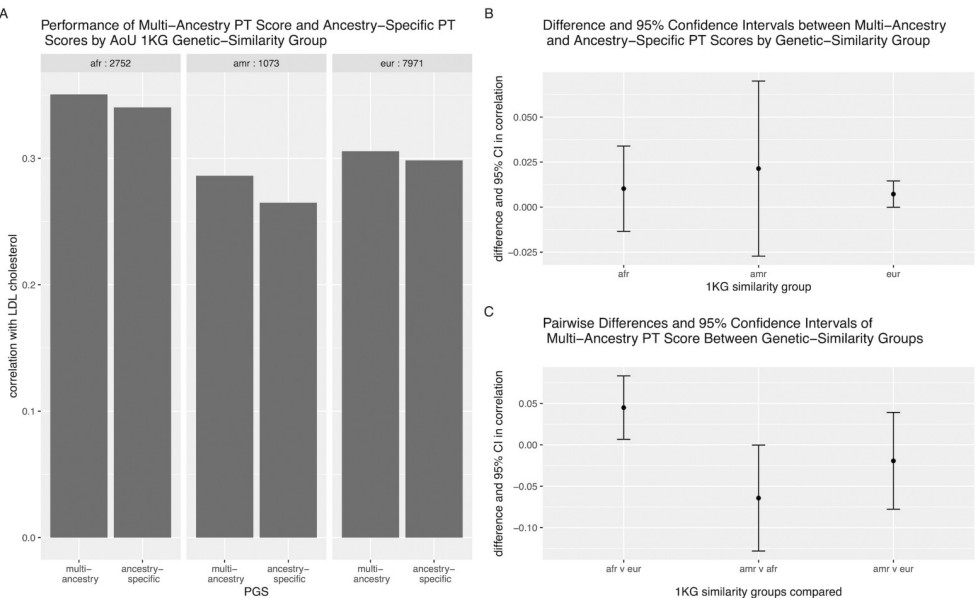

**Fig 4. Performance of multi-ancestry and ancestry-specific PGS for LDL cholesterol.** A: Bars represent correlation between PGS and inverse-rank normalized mean LDL cholesterol in each 1KG genetic-similarity group. B: Dot points represent difference in correlation between multi-ancestry and ancestry-specific PGSs and error bars indicate 95% confidence interval of this difference. C: Dot points represent difference in correlation of multi-ancestry PGS between 1KG genetic-similarity groups indicated in x-axis and error bars indicate 95% confidence interval of this difference.

with height differs across the four genetic-similarity groups, ($p_{between}$ = 1.4 × $e^{-81}$, Fig 5B), that at least one score has higher correlation with height than the others ($p_{within}$ = 6.2 × $e^{-166}$, Fig 5C), and that the pattern of score correlation with height differs across the genetic-similarity groups ($p_{interaction}$ < 1 × $e^{-300}$).

For height, the multi-ancestry score does not outperform the others in all of the 1KG genetic-similarity groups (Fig 6). We can use the Coranova 'within' hypothesis test to compare the multi-ancestry PGS to the ancestry-specific PGS in each genetic-similarity group separately. Among individuals classified as African, the score built with the African GWAS has a 0.115 higher correlation with height than multi-ancestry score, (95% CI: (0.102, 0.127), $p_{within}$ = 4.9 × $e^{-69}$). Among individuals classified as European, the score built with the European GWAS has a 0.016 higher correlation with height than the multi-ancestry score (95% CI: (0.013, 0.018), $p_{within}$ = 1.8 × $e^{-36}$). Among the other genetic-similarity groups the multi-ancestry score did outperform the corresponding ancestry PGSs. Among individuals classified as admixed American, the score built with the Hispanic GWAS has a 0.039 lower correlation with height than the multi-ancestry score (95% CI: (0.025, 0.053), $p_{within}$ = 4.5 × $e^{-08}$). Among individuals classified as East Asian, the score built with East Asian GWAS had a 0.032 lower correlation with height than the multi-ancestry score (95% CI: (0.001, 0.062), $p_{within}$ = 0.04).

## Discussion

We propose a flexible formal statistical framework for assessing the performance of two or more PGS in one or more populations with correlation, which was previously unaddressed by available methods, and provide an R package to make it easy for users to implement our proposed methods. We use simulations to evaluate our methods when applied to three PGS in two population samples and find we have well-controlled type I error and power greater than

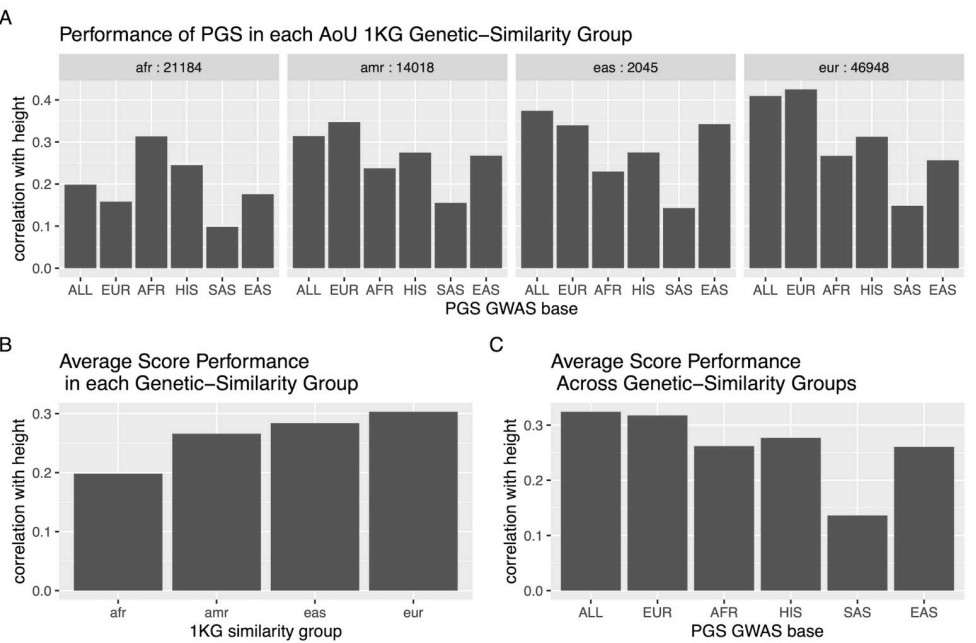

**Fig 5. Correlation of Yengo et al. PGS with inverse rank normalized mean height in each of the African, Admixed American, East Asian and European 1KG genetic-similarity groups in AoU.** A: The correlation with height for each PGS by 1KG genetic-similarity group, where each PGS is identified by the GWAS base with which it was derived. Sample size of each group specified. B: The average score performance in each AoU 1KG similarity group, corresponding to the "between hypothesis test" in which the null hypothesis is that the scores perform on average the same in the three groups. C: The performance of each PGS averaged across the AoU 1KG similarity groups, corresponding to the "within hypothesis test" in which the null hypothesis is that the scores have the same correlation with the outcome when averaged across populations.

80% for reasonable values to assess between-group and within-group differences in polygenic score performance as well as differences in the pattern of score performance across groups. Finally, we highlight the utility of our methods to adapt to researcher interests with two examples applying PGS for height and LDL cholesterol to the All of Us cohort.

The methods we propose here are uniquely appropriate for the comparison of performance of polygenic scores. Researchers generally have one of two goals when comparing polygenic scores: 1) selecting an optimal polygenic score for a specific outcome, or 2) determining an optimal polygenic score derivation procedure, comparing scores built with different methods, inputs or both. Both goals often require the comparison of multiple polygenic scores in multiple population samples. Thus, while many methods are available to compare model performance, such as those for nested and non-nested models [21], these are not generally appropriate for the needs of researchers studying polygenic scores, as they implement pairwise comparisons. Our proposed methods can be used to compare many scores at one time, as well as perform pair-wise comparisons as necessary.

The results we present in both our simulations and examples in All of Us employ parametric implementations of our proposed methods. Bilker et al. advocated for bootstrapping to derive the covariance matrix of $u$, $\hat{\Sigma}(u)$, and permutation-testing to generate p-values. However, their work was concerned with analyzing neurological exams of fewer than 75 patients, so it is not surprising that parametric methods based on asymptotic results did not work well for their purposes. We provide the option to use bootstrapping and permutation testing in our R package with the `perform_coranova_nonparametric` function, but we find that

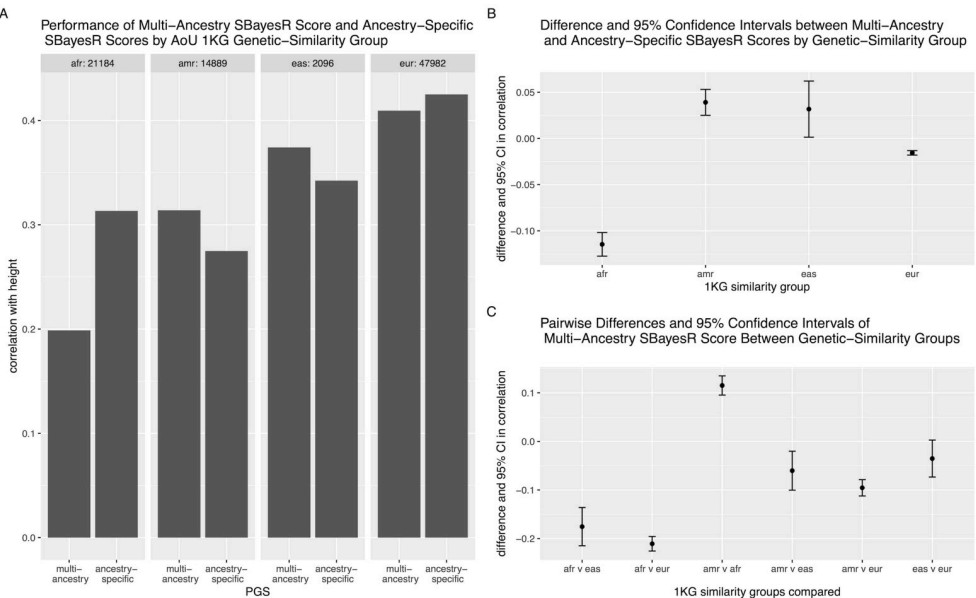

**Fig 6. Performance of multi-ancestry and ancestry-specific PGS for height.** A: Bars represent correlation between PGS and inverse-rank normalized mean Height in each 1KG genetic-similarity group. B: Dot points represent difference in correlation between multi-ancestry and ancestry-specific PGSs and error bars indicate 95% confidence interval of this difference. C: Dot points represent difference in correlation of multi-ancestry PGS between 1KG genetic-similarity groups indicated in x-axis and error bars indicate 95% confidence interval of this difference.

parametric implementations of the correlation-based hypothesis tests are appropriate when working with sample sizes of 500 or more individuals.

In our analysis of the 12 PGS developed by Graham et al. for LDL cholesterol, we apply the three Coranova hypothesis tests and first determine that the 12 PGS do not have the same correlation with LDL cholesterol when averaged across 1KG genetic-similarity groups (the "within" hypothesis), the overall the mean of the 12 PGS correlations with LDL does not differ by 1KG genetic-similarity group (the "between" hypothesis), and the pattern of correlations between the 12 PGS and the LDL outcome differs across the 1KG genetic-similarity groups ("interaction" hypothesis). In other words, we find that some PGS perform better than others, but overall, the PGS on average perform the same across the genetic-similarity groups and the pattern of the PGS performance differs by genetic-similarity group. We also use our flexible framework to identify whether we can recommend a single PGS from the 12. First, we conclude that the scores built with pruning and thresholding (PT) outperform the scores built with PRS-CS. We then compare the performance of the multi-ancestry PT PGS with the ancestry-specific PT scores in each genetic-similarity group and find that the multi-ancestry PGS does not have a significantly higher correlation with LDL cholesterol than the ancestry-specific scores. We suspect that if our sample sizes were larger we would be able to detect a significant difference, however, our results at least show that the multi-ancestry PGS performs as well as the ancestry-specific scores in each genetic-similarity group, making it an appropriate PGS to employ in all three groups. When we compare the correlation of the multi-PGS with LDL cholesterol in the three genetic-similarity groups, we find that the score performance varies significantly. It is surprising that the score is more highly correlated with LDL in the African genetic-similarity group than the European genetic-similarity group, considering individuals of African and admixed African ancestry made up only 6% of the original GWAS sample [16]. This

finding may not replicate in other cohorts with different context characteristics [22]. Still, we conclude multi-ancestry PT PGS is an optimal choice for a LDL PGS when working with a multi-ancestry population.

In contrast, in our analysis of the six PGS developed by Yengo et al. for height, we cannot make such a case for the multi-ancestry PGS, or any of the six PGS. When we apply the three Coranova hypothesis tests, we reject all three null hypotheses and find that the PGS do not have the same correlation with height, the mean correlation of the PGS with height is not consistent across the 1KG genetic-similarity groups, and that the pattern of score performance differs by genetic-similarity group. When we consider the pairwise differences between the multi-ancestry and ancestry-specific PGS in each genetic-similarity group, we find that in the AoU European and African genetic-similarity groups, the ancestry-specific PGS have a higher correlation with height than the multi-ancestry PGS, and in the AoU Admixed American and East Asian genetic-similarity groups, the reverse is true and the multi-ancestry PGS outperform the ancestry-specific scores. Thus, we cannot recommend a single score be used for all populations and recommend that the ancestry-specific PGS are used when working with individuals classified as similar to the 1KG European or African populations, and that the multi-ancestry PGS is used when working with individuals classified as similar to the 1KG admixed American or East Asian populations.

Our two examples comparing PGS performance in AoU highlight the strengths and weaknesses of our proposed approach. One of the main advantages to our proposed method is the ability to assess the performance of multiple PGS in multiple population samples simultaneously before testing more specific hypotheses. In each example, we were able to ask high-level questions about the PGS performance in multiple populations, like whether there was one score that outperformed the others and whether mean PGS correlation with the phenotype of interest differed by genetic-similarity group. Based on the results from the initial hypothesis tests, we then assessed more specific questions like whether the multi-ancestry scores outperformed the ancestry-specific scores. Our framework is highly flexible and can be employed to assess a variety of hypotheses. The hypotheses of interest will be dependent on specifics of the analysis.

However, considering the LDL cholesterol results comparing the multi-ancestry PGS to the ancestry-specific PGS, we can see our methods are sensitive to sample size. The confidence interval around the estimate of the difference between the multi-ancestry PGS and European-specific PGS is much smaller than the confidence interval around the estimate of the difference between the multi-ancestry PGS and admixed American-specific PGS (Fig 4B) because there are over seven times as many people classified as European with LDL cholesterol levels in AoU than there are people classified as admixed American with LDL cholesterol levels in AoU. Thus, while our methods can be used to assess PGS performance in different populations, if sample sizes differ, caution must be used when interpreting results.

We also advise caution when performing the between group hypothesis test when comparing many polygenic scores at once, since these tests are designed to detect a difference in mean polygenic score correlation across population groups. Additionally, when applying these tests to multiple genetic-similarity populations, we recommend that both outcome and polygenic score are adjusted for genetic principal components (PCs) with a linear model prior to analysis. This step is important to ensure that the relationship between the polygenic score and outcome is not confounded by population structure [23]. For this reason, we do not recommend utilizing this method to compare polygenic scores for disease traits, as it is not possible to adjust a binary outcome for covariates with a linear model. If researchers have reason to believe confounding by population structure is not a concern, we have implemented a nonparametric version of Coranova available with the `perform_coranova_nonparametric` function

which does not assume the data is normally distributed (see section D of S1 Appendix and section C of S2 Appendix for more details). Finally, the correlation between a polygenic score and its intended outcome is typically positive. If the correlation is not positive, this may indicate an error in designated effect allele in the polygenic score computation. We provide recommendations for implementation of these methods and substantive examples of application in our user manual available on github.

If polygenic scores are ever to be employed in a clinical setting, it is imperative that we have models that can perform well in all groups [3]. A necessity for the development of PGS for diverse populations is methodology to assess PGS performance in multiple populations. As more data becomes available, and researchers have greater access to GWAS results from varied populations, our proposed methods will be an important tool in the effort to ensure that PGS perform well in all populations.

## Supporting information

**S1 Appendix. Supplemental methods.**
(PDF)

**S2 Appendix. Supplemental results.**
(PDF)

**S3 Appendix. Supplemental figures.**
(PDF)

**S4 Appendix. R package vignettes.**
(PDF)

## Acknowledgments

We thank the Global Lipids Genetics Consortium and GIANT Consortium for making their polygenic scores publicly accessible. We also gratefully acknowledge All of Us participants for their contributions, without whom this research would not have been possible. Finally, we thank the National Institutes of Health's All of Us Research Program for making available the participant data examined in this study.

## Author Contributions

**Conceptualization:** Sophia Gunn, Kathryn L. Lunetta.

**Data curation:** Sophia Gunn.

**Formal analysis:** Sophia Gunn.

**Funding acquisition:** Sophia Gunn, Kathryn L. Lunetta.

**Investigation:** Sophia Gunn.

**Methodology:** Sophia Gunn.

**Project administration:** Sophia Gunn.

**Software:** Sophia Gunn.

**Supervision:** Kathryn L. Lunetta.

**Validation:** Sophia Gunn.

**Visualization:** Sophia Gunn.

**Writing – original draft:** Sophia Gunn.

**Writing – review & editing:** Sophia Gunn, Kathryn L. Lunetta.

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
