## [Decision Letter · Decision Letter 0]

7 Dec 2023

Dear Dr Gunn,

Thank you very much for submitting your Research Article entitled 'Correlation-Based Tests for the Formal Comparison of Polygenic Scores in Multiple Populations' to PLOS Genetics.

The manuscript was fully evaluated at the editorial level and by independent peer reviewers. Based on the reviews, we will not be able to accept this version of the manuscript, but we would be willing to review a much-revised version (although we cannot promise publication at that time). The reviewers appreciated the attention to an important problem, but raised some substantial concerns about the current manuscript. In particular, multiple reviewers felt the overall writing of the manuscript could be enhanced and that the work itself needs to be more self contained. 

Should you decide to revise the manuscript for further consideration here, your revisions should address the specific points made by each reviewer. In addition, substantial improvements are needed in the manuscript's writing to enhance clarity, enabling readers to better understand the methods, simulations, and results. We will also require a detailed list of your responses to the review comments and a description of the changes you have made in the manuscript.

If you decide to revise the manuscript for further consideration at PLOS Genetics, please aim to resubmit within the next 60 days, unless it will take extra time to address the concerns of the reviewers, in which case we would appreciate an expected resubmission date by email to plosgenetics@plos.org.

We are sorry that we cannot be more positive about your manuscript at this stage. Please do not hesitate to contact us if you have any concerns or questions.

Yours sincerely,

Xiang Zhou, Ph.D.

Academic Editor

PLOS Genetics

Michael Epstein

Section Editor

PLOS Genetics

Reviewer's Responses to Questions

**Comments to the Authors:**

Reviewer #1: Correlation-Based Tests for the Formal Comparison of Polygenic Scores in Multiple Populations

Gunn and Lunetta

This study introduces correlation-based methods to assess the performance of Polygenic Scores (PGS) by building upon prior work that developed a statistical framework and robust test statistics for comparing multiple correlation measures across diverse populations. The adaptable framework presented here can be extended to a broader range of hypothesis tests compared to existing methods. The authors validate the proposed methods through simulations and illustrate their utility with two examples: evaluating previously developed PGS for low-density lipoprotein cholesterol and height across multiple populations in the All of Us cohort. Additionally, the authors have created an R package called 'coranova' featuring both parametric and non-parametric implementations of the described methods.

This study is interesting, however, there are comments and questions that should be carefully considered by the authors.

1. This study employs a correlation-based test as a fundamental aspect of its methodology. It may be essential for the authors to explicitly articulate the distinction between correlation-based and R^2-based tests, as these are two distinct test statistics with unique properties. I strongly recommend that the authors delve deeper into this comparison, specifically exploring their performance in non-nested model comparisons. For instance, a valuable approach would be to compare the performance of correlation-based and R^2-based tests using Vuong’s test as a benchmark for non-nested model comparisons.

Vuong, Quang H. (1989). "Likelihood Ratio Tests for Model Selection and non-nested Hypotheses" (PDF). Econometrica. 57 (2): 307–333.

For nested model comparisons, a parallel analysis is recommended, with a focus on assessing the performance of correlation-based and R^2-based tests using the likelihood ratio test as a reference. Given the availability of these methods in R-packages, conducting such comparisons should be straightforward.

I think that the R^2-based test primarily serves to assess model fit, akin to the likelihood ratio test for nested model comparisons and Vuong’s test for non-nested model comparisons. However, correlation-based test is slightly different from R^2-based or likelihood ratio test including Vuong’s.

To enhance clarity for readers, it is advisable for the authors to explicitly elucidate this distinction in the Introduction or early section of the Results, ensuring a clear understanding of the metrics employed in their study.

2. The current study does not address nested model comparisons. Is there a specific rationale for this omission? I recognize that dealing with nested model comparisons can become intricate, especially when multiple predictors are involved. However, it is noteworthy that such comparisons can be readily applied to simpler pairwise scenarios.

Nested model comparison holds significance as it allows for the hypothesis that the inclusion of an additional predictor significantly enhances the model fit. It would be beneficial for the authors to contemplate how this methodology can be effectively employed, particularly when dealing with multiple predictors. Considering and discussing the implications of nested model comparisons in the context of multiple predictors would enhance the completeness and robustness of the study.

3. In the abstract, the phrase 'the comparison of multiple correlation measures in multiple populations, perhaps correlated' may be misleading. The term 'correlated' should be omitted, as individuals across populations are assumed to be independent of each other.

4. The deflated type I error rate is evident in the R2-based test, as illustrated in Figure 1, particularly when tau is low. This observation aligns with expectations, as lower tau values lead to sign switches in some replicates for the two sets of estimated taus, attributable to sampling errors. Consequently, the discrepancy between two Polygenic Scores (PGSs) for R^2-based metrics is smaller compared to the correlation-based test. It is noteworthy that, with a larger sample size, the deflated type I error rate tends to be mitigated. To further substantiate this, the authors may consider verification with a substantial sample size, such as n=10,000.

Additionally, it is crucial to note that the test outcome depends on the formulated hypothesis. Specifically, the hypothesis should assert no difference between the two sets of R^2 when the direction of association is consistent between the two models. In cases where the direction of association differs, the R^2 for negative regression coefficients should manifest as negative when subjected to subtraction. The authors are encouraged to consider and address this nuance in their analysis.

5. In Figure 4, I suggest the authors scrutinize why the test for 'between' decreases as phi increases. Additionally, it would be beneficial to discuss any insights into why the tests for other categories increase with an increasing phi, unless such explanations have already been provided in the text.

6. Figures 5 and 7 could be significantly enhanced by emphasizing the hypothesis related to multiple PGSs across populations (as depicted in section 3.2). Additionally, it is essential to improve the comprehensibility of the legends associated with these figures.

7. When comparing European and African datasets using the GitHub page's function (perform_coranova_parametric(list(afr, eur), "pheno", c("pgs1", "pgs2", "pgs3"))), the author conducted a non-nested model comparison between two independent ancestries, where the sample size can differ between the two populations. Specifically, the author compared the following models:

• Model 1: afr$Pheno ~ afr$pgs1 + afr$pgs2 + afr$pgs3

• Model 2: eur$Pheno ~ eur$pgs1 + eur$pgs2 + eur$pgs3

Is my understanding correct? Nevertheless, it is challenging to discern how the number of Polygenic Scores (PGSs) varies in each population. Additionally, the process may not be very user-friendly, especially if the headers differ across data frames. It is recommended that the authors develop a user-friendly manual and an interface within the functions to enhance usability.

Reviewer #2: Dr. Gunn and colleagues have introduced a correlation-based approach to test for differences in polygenic scores (PGS) across multiple populations. Utilizing the asymptotic covariance results of correlations from Olkin & Finn, they have developed a test framework and applied it to data from the All of Us project, assessing traits like LDL and height. While the framework is compelling, there are aspects of the manuscript that could be refined for clarity and precision. I suggest the following revisions:

1. On page 3 section 3.1, the definition of \\mu as "the vector of population values" should be clarified to represent the population mean of sample correlations. Moreover, the hypothesis testing section should be made more formally precise by presenting the null hypothesis without the term "mean". If u is defined within a specific population j, why don’t you define u as u_j=(r_(1,j),…,r_(P,j)) and μ_j=(ρ_(1,j),…,ρ_(P,j) ). Then the “between” test null hypothesis can be written as H_0:μ_1=⋯=μ_K. Please update your hypothesis testing section to make it clear.

2. Figure 1 mentions a parameter \\delta that is absent from the figure. The figure should be updated to include this parameter, or the text should be amended to reflect its contents accurately. Additionally, the figure's legend needs more detailed explanations, especially regarding the null hypothesis tested. The same applies to Figure 2, where the legend should provide sufficient context for the results presented.

3. An explanation is needed for why the R2-based test shows significantly less power when the correlation between PGS and outcome is low (e.g., tau = 0.05).

4. Figure 2 is comparing the power of Coranova and R2 Redux. Figure 2 is referenced in the following sentence which focuses on type I error: “We find that our correlation-based method has well controlled type I error when assessing differences in two polygenic scores in a single population (figure 1 and 2)”. It would be helpful to describe the results of figure 2 separately.

5. I am a little bit confused with the setting of Figure 4. Why do you want to conduct the test for “interaction” and “within (difference across scores)” in the setting of “testing between hypothesis”

6. Additional clarification is needed for the results depicted in Figure 4, where the power for detecting differences across PGS is higher when the PGS are more correlated with the outcome (high tau). This seems unintuitive, as it suggests that differences should be harder to discern when all PGSs are highly correlated with the outcome.

7. The manuscript would benefit from a more detailed description of the simulation settings. Each setting ("between," "within," "interaction") should have clear parameters defined against which the null hypothesis is tested. The current descriptions in the figure legends are brief.

8. In the third paragraph of discussion, I am confused with the following sentence: ”12 PGS do not have the same correlation with LDLD cholesterol, mean PGS correlation with LDL does not differ by 1KG genetic-similarity group, and the pattern of correlation between the PGS correlation between the PGS and LDL cholesterol differs by genetic similarity group”. It would be helpful if you could clarify it a little bit more.

9. The modeling of correlations and the corresponding covariance matrix in your framework shows admirable flexibility, particularly with the potential to use various contrast matrices A for hypothesis testing. The use of a global test to discern differences in the correlation of PRS across ancestries is insightful. However, the true robustness of the approach may be challenged when applied to scenarios involving more than two PRS methods, as is common in practice. In such cases, the global test might encounter limitations due to an increased degrees of freedom penalty, which could impact its power. It would be compelling to see the framework applied to these more complex scenarios. Specifically, it would be valuable if the framework could address detailed questions in a multi-ancestry context, such as: 1) Are there performance differences across ancestries for a given PRS method? 2) Within each ancestry, which PRS method provides the best prediction performance? 3) Considering all ancestries, which PRS method emerges as the most effective?

10. When dealing with two PGSs derived from different sources, there might be instances of inverse correlations with similar magnitudes due to the use of different effect alleles. I’m curious whether the Coranova package can account for such scenarios, which are not typically an issue in R2 evaluations but are pertinent to correlation analyses.

11. It would be interesting to see an example wi

---

## [Decision Letter · Decision Letter 1]

22 Feb 2024

Dear Dr Gunn,

Thank you very much for submitting your Research Article entitled 'Correlation-Based Tests for the Formal Comparison of Polygenic Scores in Multiple Populations' to PLOS Genetics.

The manuscript was fully evaluated at the editorial level and by independent peer reviewers. There are a few remaining comments from the reviewers that we ask you address in a revised manuscript.

We therefore ask you to modify the manuscript according to the review recommendations. Your revisions should address the specific points made by each reviewer.

Yours sincerely,

Xiang Zhou, Ph.D.

Academic Editor

PLOS Genetics

Michael Epstein

Section Editor

PLOS Genetics

Reviewer's Responses to Questions

**Comments to the Authors:**

Reviewer #1: In the context of a polygenic score and its associated outcome, the correlation is typically expected to be positive. However, it's essential to note that there can be instances where a negative correlation arises due to factors such as sampling error or a negative association between estimated SNP effects and the outcome.

Please revise 'always' to 'typically' at lines 83 and 441. Other than that, I have no further comments

Reviewer #2: The authors have fully addressed my comments.

Reviewer #3: The authors have addressed my major concerns. I have the following comments.

1. In the introduction section: .... this correlation should always be positive, ranging from 0 (the PGS is completely independent of

84 its outcome), to 1 (the PGS is perfectly linearly associated with its outcome). This statement requires more justification.

2. In the responses to the reviewers report, please see the response to comment 3 from the reviewer 3. In the portion where the authors have mentioned contrast matrix using matrix algebra, the columns of the first contrast matrix are not linearly independent. We can add the 1st and second columns to get the last column. Or, did the authors mean the row vectors? Because R2 can not have a basis containing more than 2 linearly independent vectors.

**Have all data underlying the figures and results presented in the manuscript been provided?**

Reviewer #1: Yes

Reviewer #2: None

Reviewer #3: Yes

PLOS authors have the option to publish the peer review history of their article (what does this mean?). If published, this will include your full peer review and any attached files.

Reviewer #1: No

Reviewer #2: **Yes: **Haoyu Zhang

Reviewer #3: No

---

## [Editor Report · Decision Letter 2]

3 Apr 2024

Dear Dr Gunn,

We are pleased to inform you that your manuscript entitled "Correlation-Based Tests for the Formal Comparison of Polygenic Scores in Multiple Populations" has been editorially accepted for publication in PLOS Genetics. Congratulations!

Yours sincerely,

Xiang Zhou, Ph.D.

Academic Editor

PLOS Genetics

Michael Epstein

Section Editor

PLOS Genetics

Comments from the reviewers (if applicable):

**Data Deposition**

http://datadryad.org/submit?journalID=pgenetics&manu=PGENETICS-D-23-01188R2

**Press Queries**

---

## [Editor Report · Acceptance letter]

18 Apr 2024

PGENETICS-D-23-01188R2 

Correlation-Based Tests for the Formal Comparison of Polygenic Scores in Multiple Populations 

Dear Dr Gunn, 

We are pleased to inform you that your manuscript entitled "Correlation-Based Tests for the Formal Comparison of Polygenic Scores in Multiple Populations" has been formally accepted for publication in PLOS Genetics! Your manuscript is now with our production department and you will be notified of the publication date in due course.

With kind regards,

Livia Horvath

PLOS Genetics

On behalf of:
